# *Poa annua*: An annual species?

**Devon E. Carroll**[1], **Brandon J. Horvath**[1], **Michael Prorock**[2], **Robert N. Trigiano**[3], **Avat Shekoofa**[1], **Thomas C. Mueller**[1], **James T. Brosnan**[1]*

**1** Department of Plant Sciences, The University of Tennessee, Knoxville, Tennessee, United States of America, **2** Mesur.io, Yanceyville, North Carolina, United States of America, **3** Department of Entomology and Plant Pathology, The University of Tennessee, Knoxville, Tennessee, United States of America

* jbrosnan@utk.edu

## Abstract

As the Latin name *annua* implies, the species *Poa annua* L. is thought to have an annual life cycle. Yet, there are many reports in literature of *P. annua* persisting as a perennial. Considering that *P. annua* senescence patterns do not align with other true annual species, we hypothesized that *P. annua* is similar to other perennial, $C_3$ turfgrass species that are subject to a confluence of environmental factors that can cause mortality. Four experiments were conducted in Knoxville, TN with the objective of determining environmental factors lethal to *P. annua*. A field monitoring study assessed 100 *P. annua* plants across ten grassland micro-environments from May to October 2020. Forty plants survived the summer and confirmed the existence of perennial *P. annua* ecotypes. Analysis of environmental factors at the time of plant death indicated soil moisture, soil temperature, and pathogenic infection were associated with mortality. A series of glasshouse or field experiments were conducted to investigate the effects of each factor on *P. annua* mortality. Soil moisture and soil temperature were not lethal to *P. annua* in the glasshouse, except under extreme conditions not typical in the field. A field study assessed mortality of plants from pathogenic infection and indicated that *P. annua* plants treated with fungicide throughout the summer survived year-round, whereas plants not receiving fungicide applications senesced. These findings support our hypothesis that *P. annua* is of a perennial life cycle, which can be influenced by environmental conditions. We suggest that the name *P. annua* is likely a misnomer based on its modern interpretation.

## Introduction

*Poa annua* L. is a $C_3$ grass species and a common component of urban grasslands. The ubiquity of *P. annua* in terms of both distribution and abundance has been compared to that of *Homo sapiens* [1]. Although the species can be desirable, it is also recognized as the most troublesome weed of urban grasslands and second most troublesome weed of all grass crops [2].

An annual species is defined by abrupt senescence following completion of a single reproductive cycle, even if growing conditions are optimal [3]. Grass crops such as *Triticum spp.* (wheat) and *Zea mays* L. (maize) are annuals that senesce after fruiting regardless of environmental surroundings [4, 5]. Despite its name meaning annual in Latin, *P. annua* does not

**Data Availability Statement:** Data are available for experiments 1 and 4 at https://colab.research.google.com/drive/16eIDgyFMX08K2szhc_LdXmbYavf982jY?usp=sharing. Data are available for experiments 2 and 3 at https://mesurpubshare.

blob.core.windows.net/public/PoaLifeCycle_PLOS_
Data.xlsx.

**Funding:** The author(s) received no specific
funding for this work.

**Competing interests:** The authors have declared
that no competing interests exist.

follow the same pattern of absolute monocarpic senescence. Furthermore, reports of *P. annua* surviving as a perennial are abundant [6–30].

It is unclear why *P. annua* is considered an annual species. Carl von Linné (Carl Linnaeus) provided no criteria when naming the species; in 1753 he describes *P. annua* as, "Extended corn bluegrass with straight angles, with smoothed spikes, with a compressed, slanted top. Minor field tuberous grass. Greater red field tuberous grass. It lives on the European paths." and in 1754 states only that he identified the plant in a field in Denmark [31, 32]. Perhaps the epithet *annua* is taken too literally in modern times and was originally intended to describe an annual event such as inflorescence production.

Because *P. annua* is known to copiously produce seed [13, 14, 16, 33], the *annua* epithet may be related to observations of inflorescence characteristics. While high fecundity is frequently associated with an annual life cycle [17, 34, 35] other species common to urban grasslands such as *Taraxacum offiicinale* Weber ex. Wiggers (dandelion) and *Trifolium repens* L. (white clover) are prolific in inflorescence/seed production and survive perennially [36, 37]. Additionally, *P. annua* can produce seed continuously [7, 17, 19] resulting in classification as a polycarpic perennial [9, 14, 38]. By definition, a polycarpic species cannot be annual as seed is produced more than once and is not followed by plant death [3].

Could *P. annua* be a perennial species? A large-scale study of nearly 7,000 *P. annua* plants harvested across Europe (from Portugal to Finland) showed the majority of plants (5,352; 79.3%) were perennial, whereas only 1,398 (20.7%) survived less than one year [9]. This response aligns with theories of senescence and selection, which favor a perennial over annual life cycle [39–41]. For example, long-lived individuals produce more offspring over time than those with short lifespans, causing natural selection to advance toward perenniality [14, 39]. Therefore, reproductive characteristics indicate a perennial life cycle is superior for global establishment. It is more likely *P. annua* is a perennial species, given that plants have successfully colonized all seven continents across a wide range of environments [42] in potentially as few as 10,000 years [7, 43, 44].

Therefore, similar to Johnson [45], we hypothesize that *P. annua* is comparable to other perennial, $C_3$ urban grassland species, which are subject to a confluence of environmental factors in summer that can result in mortality. Premature senescence, particularly due to pathogenic infection, of perennial plants in summer months after seed production may give the appearance of an annual life cycle. Such an occurrence is similar in crops of *Solanum lycopersicum* L. (tomato), *S. tuberosum* L. (potato), and *Gossypium hirsutum* L. (cotton), which are perennial species that are cultivated as annuals due to premature senescence in climates with adverse temperatures [46–48].

While scientists have reported *P. annua* is highly sensitive to heat and drought stress [8, 9, 11, 13, 15, 19, 45, 49–51], the influence of these factors has often been assessed individually, for short periods of time, and in greenhouses or growth chambers. In the field, fluctuations in temperature and moisture over time can affect disease incidence in susceptible hosts such as *P. annua* when pathogens are present.

*Poa annua* is susceptible to numerous diseases including dollar spot (*Clarireedia* spp.), anthracnose (*Colletotrichum cereale*), Pythium blight (*Pythium* spp.), and brown patch (*Rhizoctonia solani*) [52]. Pathology literature describes these diseases as most devastating, resulting in summer decline or plant death, when plant stress is induced via anthropogenic activities or unfavorable environmental conditions [53–56]. Pathogenicity of the aforementioned diseases peaks with elevated air temperature and atmospheric humidity [54, 57–59]. Therefore, stressful environmental conditions (e.g., elevated air temperature, drought, etc.) may falsely appear to cause *P. annua* senescence when heightened pathogenetic activity is the actual cause of mortality.

Although the nature of *P. annua* transience in some situations remains uncertain, review of literature indicates the species may be incorrectly botanically characterized [60]. If any environmental factors are causing plant death rather than programmed senescence after inflorescence production, classification as an annual species does not follow the accepted definition of such a life cycle [3]. Given the world-wide presence of the species and difficulties related to management [2, 61], efforts to better understand *P. annua* life cycle are warranted. Therefore, the objective of this study was to determine environmental factors lethal to *P. annua*.

## Materials and methods

### Ethics statement

Field research and sample harvesting was conducted on university property, Three Ridges Golf Course, or Oak Ridge Country Club. No permits were required for testing or plant collection. Permission for use was granted by site managers.

### Life cycle observation

An observational field study was initiated on 11 May 2020 at three locations in Knoxville, TN (S1 Table). One hundred *P. annua* plants were monitored throughout the summer and fall until 27 Oct. 2020. Monitoring occurred in ten unique micro-environments across the three study locations (Table 1). Experimental areas within micro-environments measured 1 x 2 m. Ten *P. annua* plants were identified based on their similar size and growth stage and monitored within each area. Metal rings (7.6 cm in diameter and 2.8 cm in length) were installed around each plant to allow observation of the same *P. annua* plants throughout the study. Herbicides were not applied to the experimental areas during the observational period.

At study initiation and every three weeks thereafter, photographs were taken of each *P. annua* plant. Plant mortality was also visually recorded on each rating date and scored as a lack of green tissue aboveground within each metal ring. A multifactor F-score analysis of climatic data related to death events was conducted using Python (Version 3.9.9, https://www.python.org/) programming language. Libraries used in analyses included 'pandas' (v. 1.3.4; https://pandas.pydata.org/) for data handling, 'NumPy' (v. 1.22.0; https://numpy.org/) for numerical transformations and math, 'scikit learn' (v. 1.0.1; https://scikit-learn.org/stable/) for modeling and related functions, 'SciPy' (v. 1.7.1; https://scipy.org/) for code modeling and curve fitting, 'matplotlib' (v. 3.5.0; https://matplotlib.org/) and seaborn (v. 0.11.2;

**Table 1. Site characteristics for micro-environments monitored from 11 May to 27 October of 2020 to assess *Poa annua* survival.**

| Number[†] | GPS coordinates | Height of cut (cm) | Soil series | Soil texture | Soil pH | Soil organic matter (%) |
|---|---|---|---|---|---|---|
| 1 | 36.091N, -83.845W | 1.3 | Urban land-Udorthents complex | Loam (42% sand, 46% silt, 12% clay) | 5.5 | 5.6 |
| 2 | 36.091N, -83.845W | 1.3 | Urban land-Udorthents complex | Sand (90% sand, 4% silt, 6% clay) | 6.1 | 2.3 |
| 3 | 36.091N, -83.845W | 1.3 | Urban land-Udorthents complex | Sandy loam (76% sand, 16% silt, 8% clay) | 5.9 | 3.5 |
| 4 | 36.091N, -83.845W | 1.3 | Urban land-Udorthents complex | Loam (40% sand, 46% silt, 14% clay) | 5.4 | 7.9 |
| 5 | 36.091N, -83.845W | 1.3 | Urban land-Udorthents complex | Silt loam (32% sand, 58% silt, 10% clay) | 5.8 | 7.3 |
| 6 | 36.091N, -83.845W | 1.3 | Urban land-Udorthents complex | Sand (92% sand, 6% silt, 2% clay) | 6.1 | 2.9 |
| 7 | 35.911N, -83.955W | 1.6 | Waynesboro loam | Silt loam (18% sand, 60% silt, 22% clay) | 6.4 | 6.4 |
| 8 | 35.980N, -84.324W | 1.4 | Collegedale silt loam | Loamy sand (82% sand, 14% silt, 4% clay) | 6.8 | 5.1 |
| 9 | 35.980N, -84.324W | 6.4 | Collegedale silt loam | Silt loam (40% sand, 50% silt, 10% clay) | 7.2 | 4.2 |
| 10 | 35.980N, -84.324W | 1.4 | Collegedale silt loam | Loamy sand (84% sand, 14% silt, 2% clay) | 6.6 | 3.4 |

[†] Micro-environments are assigned a number for clarification of results.

https://seaborn.pydata.org/) for visualization, and 'xgboost' (v. 1.5.1; https://xgboost.readthedocs.io/en/stable/) for broad modeling to determine feature importance. Climatic data were obtained through mesur.io (Yanceyville, NC).

## Soil temperature evaluation

A glasshouse experiment was conducted from 24 Feb to 14 April 2021 in Knoxville, TN (35.946590, -83.939360) to determine the effects of soil temperature on *P. annua* mortality. Two experimental runs were conducted concurrently in adjacent glasshouse bays.

Ecotypes from micro-environments one and two (Table 1) in the aforementioned observational study were selected for inclusion on the basis of observed length. Mature *P. annua* plants were harvested from the field on 24 Feb. 2021. Harvest location one was comprised of plants that senesced during 2020, whereas harvest location two contained plants that survived the entire duration of our observational experiment in 2020. Plants were harvested within a 7.5 m$^2$ area as undisturbed soil cores (10.2 cm diameter) containing aboveground biomass and an intact root system. Plants of similar size, tiller number, and growth stage (e.g., no inflorescence) were selected for collection. Extracted soil cores were cut to a uniform depth of 7.6 cm, such that no roots were visibly protruding.

**Experimental setup and design.** Soil cores containing plants were immediately transferred to a glasshouse and installed in water baths devoid of water at the time of installation. Water baths were constructed in advance of plant harvest by mounting polyvinyl chloride (PVC) pipes (10.2 cm in length and diameter) in plastic containers (51 x 39 x 15 cm) (Fig 1). Pipes were affixed to the container using waterproof caulk atop drainage holes. Three *P. annua* plants of each ecotype were installed in pipes by sliding the core into the pipe without

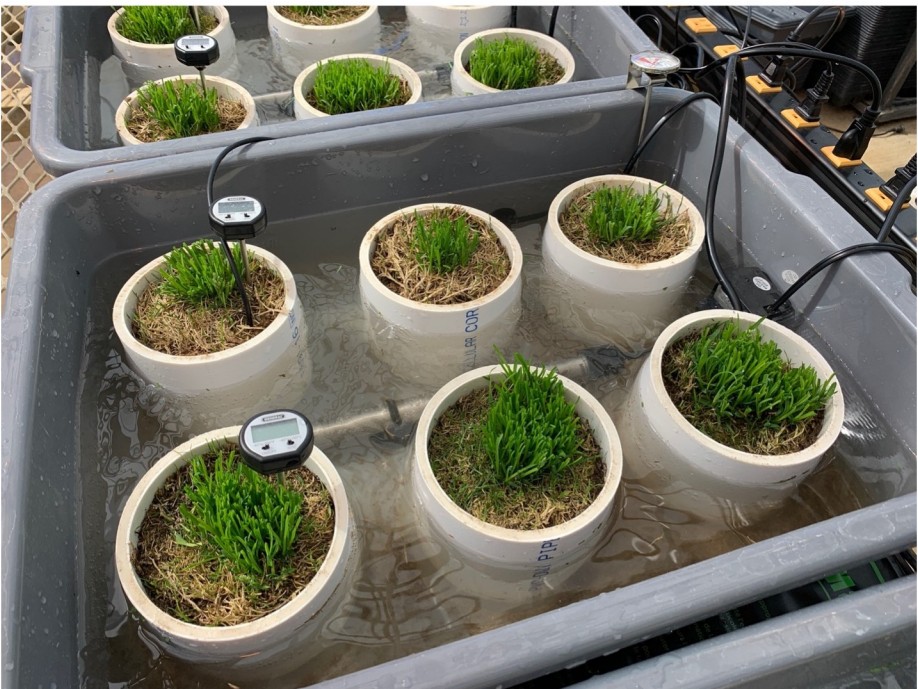

**Fig 1. *Poa annua* plants harvested on 24 February 2021 in Knoxville, TN placed in polyvinyl chloride pipes (10.2 cm diameter and length) and mounted in plastic tubs.** Water added outside of pipes was heated to 21.1, 26.7, 32.2, or 37.8˚C to elevate soil temperatures.

disturbance. Pipes were randomized within each water bath. Filter paper was placed in the bottom of each tube to prevent loss of soil while allowing for water movement.

Once installed in water baths, plants were supplied with complete fertilizer (20N-20P$_2$O$_5$-20K$_2$O; Southern Ag Triple Twenty with minors; Hendersonville, NC) to deliver 49 kg N ha$^{-1}$ and immediately irrigated (5 mm) via an overhead sprinkler system.

Plants were acclimated in water bath systems under glasshouse conditions for seven days (24 Feb to 3 March 2021) before treatments were imposed. During this period, glasshouse conditions averaged 50% relative humidity and 22.8˚C air temperature. A 16/8 hour day/night artificial photoperiod was imposed via metal halide lights (1000 W; P.L Light Products, Beamsville, ON, Canada) with night occurring from 2200 to 0600 hours. Five mm of irrigation was applied daily between 0700 and 0900 hours. Plants were clipped to a height of 2.5 cm by hand using scissors on days three and seven (final) of acclimation.

After seven days of acclimation, plants were subjected to one of four soil temperatures imposed by water baths for four weeks (3 to 31 March 2021). The method of using water baths to impose variable soil temperatures is similar to those used in other experiments investigating soil temperature effects on grass mortality [62, 63]. Water baths were arranged in a split-plot design with three replications. The whole plot factor was soil temperature whereas the split-plot factor was *P. annua* ecotype (n = 72 plants; 2 ecotypes x 3 plant replications x 4 soil temperatures x 3 water bath replications).

Aquarium heaters with digital thermometers (500 W; Hygger, Renton, WA) were used to manipulate water temperature in each bath to augment soil temperature. Each water bath was also equipped with a 4 W circulation pump to prevent algal infestation and a secondary liquid thermometer to monitor water temperature. Heat from water in baths was transferred to soil cores containing *P. annua* until soil temperature was in equilibrium with water temperature.

For the duration of this experiment, water bath temperatures were 21.1, 26.7, 32.2, or 37.8˚C. Temperatures in this range were chosen because previous studies assessing soil temperature effects on C$_3$ grasses (such as *P. annua*) consider 20˚C and 35˚C to be optimal and supraoptimal soil temperatures, respectively, for root growth [62, 63]. For water baths set to 26.7, 32.2, or 37.8˚C, water was added outside of PVC pipes containing *P. annua* plants to a depth of 7.6 cm. Water baths set to these temperatures were placed on a heating mat (iPower, Irwindale, CA) to prevent heat loss and ensure intended temperatures were achieved. At treatment initiation, aquarium heaters were set to the desired temperature. Within 12 hours of heat imposition, soil temperatures were in equilibrium with the desired water temperature. The 21.1˚C temperature was achieved using the same equipment devoid of water.

Soil temperatures were monitored in soil cores within one temperature replication of each water bath using external soil sensors (#3667; Spectrum Technologies Inc., Aurora, IL) programmed to record soil temperature every 15 min and transmitted to a micro station (Watch-Dog 1000 Series; Spectrum Technologies Inc., Aurora, IL). Micro stations also recorded air temperature and humidity every 15 minutes. Secondary digital soil thermometers (#6300; Spectrum Technologies Inc., Aurora, IL) were monitored daily in one replication of each water bath as well. In experimental run one, mean air temperature was 22.6˚C and mean relative humidity was 49.4%. In experimental run two, mean air temperature was 22.5˚C and mean relative humidity was 46.0%.

Soil temperature treatments were imposed for four weeks. During the treatment period, plants were clipped daily to a height of 2.5 cm by hand using scissors and water baths were refilled to the appropriate water level as needed. Plants were maintained at a constant gravimetric moisture content via addition of supplemental water (15 to 23 mL) twice per week. After four weeks of exposure to soil temperature treatments, plants were subjected to ambient soil temperature for two weeks (31 March to 14 April 2021) to assess the potential for

regrowth. During this time, plants were not clipped, and supplemental irrigation (5 mm) was applied daily via an overhead misting system.

**Data collection and analysis.**  Plants were visually evaluated daily for mortality, rated as all aboveground tissue necrotic, with death confirmed via assessments of photochemical efficiency using a handheld fluorometer (Fv/Fm meter; Opti-Sciences, Inc., Hudson, NH) to measure Fv/Fm ratio. Two readings were taken from each plant once per week after randomly placing five- to- ten attached leaves into dark adaption chambers for 20 minutes. Plants were considered dead when the Fv/Fm ratio was zero. For most plant species, Fv/Fm ratios in the range of 0.79 to 0.84 are considered optimum, whereas lower values indicate plant stress [64, 65].

Data were subjected to analysis of variance using the SAS (University Edition, SAS, Cary, NC) mixed procedure. Fixed effects included experimental run, soil temperature, and *P. annua* ecotype, while water bath (block) was considered a random effect. Therefore, experimental run, soil temperature, and *P. annua* ecotype interactions were tested. Analysis of variance revealed experimental run and *P. annua* ecotype were not significant factors for photochemical efficiency data. Figures were generated in Prism (Prism 9 for Mac, Graph-Pad software, La Jolla, CA).

## Soil water evaluation

A glasshouse experiment was conducted from 21 April to 9 June 2021 in Knoxville, TN (35.946590, -83.939360) to explore the effects of soil moisture on *P. annua* mortality at different soil temperatures. Two experimental runs were conducted simultaneously in separate glasshouse bays.

Because no significant ecotype interaction was identified in the soil temperature evaluation experiment, *P. annua* plants for this experiment were harvested from a single location to limit the number of experimental variables. On 21 April 2021, mature *P. annua* plants were harvested from micro-environment two (Table 1), where plants survived the entire experiment in 2020. Harvest methods were identical to those previously described. All harvested plants were of similar size and had inflorescences present at the time of collection. The collection area was treated with cyazofamid (Segway; PBI Gordon Corporation, Shawnee, KS) at 1.1 kg ha$^{-1}$ 24 hours prior to harvest to prevent confounding effects of soil pathogens. Fungicide was immediately irrigated into the soil after application.

**Experimental setup and design.**  After harvest, soil cores containing *P. annua* plants were immediately transferred to a glasshouse. Soil cores were tightly wrapped with polyethylene plastic (0.15 mm thick) to prevent water loss via evaporation. *Poa annua* foliage was carefully moved to protrude out of a 2.5 cm incision at the top of each core (Fig 2). Four drainage holes were created in the bottom of soil cores. Plastic wrapped soil cores were installed in PVC pipes (10.2 cm in length and diameter).

Once installed in PVC pipes, plants underwent 7 d of acclimation (21 to 28 April 2021) in a glasshouse under conditions described previously for photoperiod, nutrient application, irrigation, and clipping. Ambient conditions during acclimation averaged 64.9% relative humidity and 22.2˚C air temperature in experimental run one and 53.9% relative humidity and 22.1˚C air temperature in experimental run two.

At the termination of the acclimation period, soil cores were saturated to the point of water freely draining through drainage holes. Water was allowed to drain overnight, resulting in soil cores at pot capacity at the onset of treatment; an approach used by other researchers exploring effects of soil water stress on plants [66]. Thirty-six *P. annua* plants in each experimental run were subjected to a 2 x 2 factorial arrangement of soil temperature and irrigation regime for

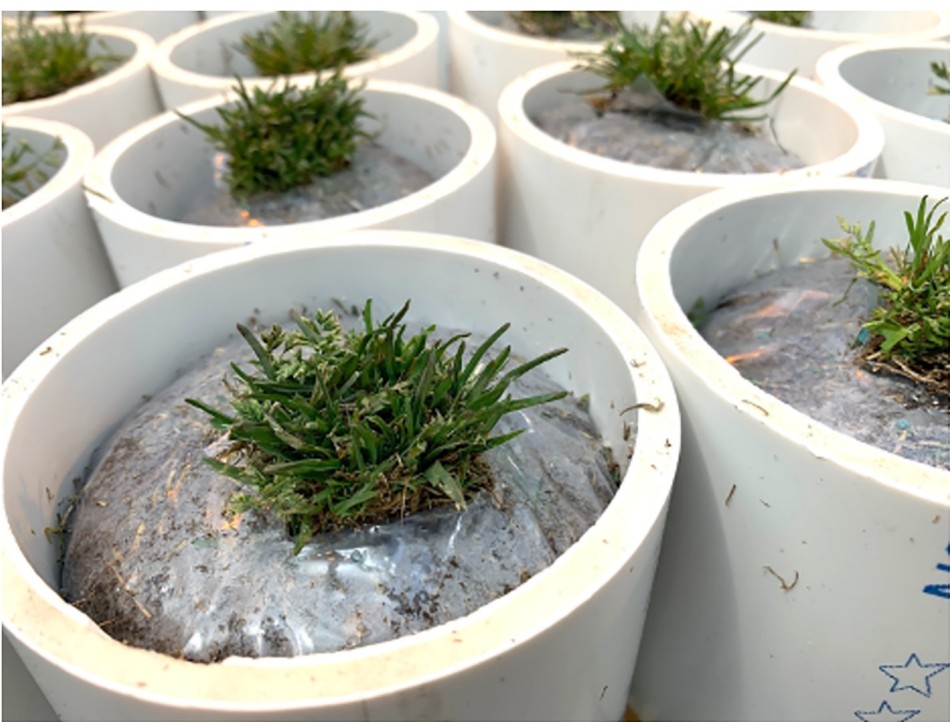

**Fig 2. *Poa annua* plants harvested in soil cores (10.2 cm diameter) wrapped in plastic (0.15 mm) and installed in polyvinyl chloride pipes (10.2 cm diameter and length).**

six weeks (28 April to 9 June 2021). During treatment, plants received no supplemental nutrients and were clipped daily to a 2.5 cm height using scissors.

Soil temperature was either ambient or elevated via a heating mat (iPower, Irwindale, CA). Irrigation treatments included well-watered (n = 12) or full drought (n = 24; Fig 3). Irrigation needs were determined by calculating daily plant transpiration [66]. Individual PVC pipes were weighed at 1200 to 1300 hours daily. Daily transpiration was calculated as the difference in weight of each pot on successive days. The well-watered plants were irrigated by hand daily to maintain a weight of no less than 50 g of their initial weight. A syringe was inserted through the incision at the top of the core to facilitate controlled watering. Plants subjected to full drought conditions received no supplemental irrigation throughout the entirety of the study.

Air temperature, relative humidity, and soil temperature were monitored as previously described. In experimental run one, soil temperature averaged 26.8˚C in cores subjected to ambient conditions and 34.6˚C in cores subjected to elevated soil temperature. Mean air temperature was 26.6˚C and mean relative humidity was 59.4%. In experimental run two, mean soil temperatures were 24.7˚C and 32.0˚C for cores subjected to ambient and elevated soil temperature, respectively. Mean air temperature was 25.7˚C and mean relative humidity was 56.1.%.

**Data collection and analysis.** Data collection and analyses were similar to those described in the soil temperature experiment. Plants were evaluated daily for mortality on a binary scale where 0 = alive (green tissue observed) and 1 = dead (all tissue necrotic). Mortality was confirmed via photochemical efficiency readings. *Poa annua* photochemical efficiency was evaluated three times per week by measuring chlorophyll fluorescence using a handheld fluorometer (Fv/Fm meter; Opti-Sciences, Inc., Hudson, NH). Two readings were taken from each plant after randomly placing five- to- ten attached leaves into dark adaption chambers for 20

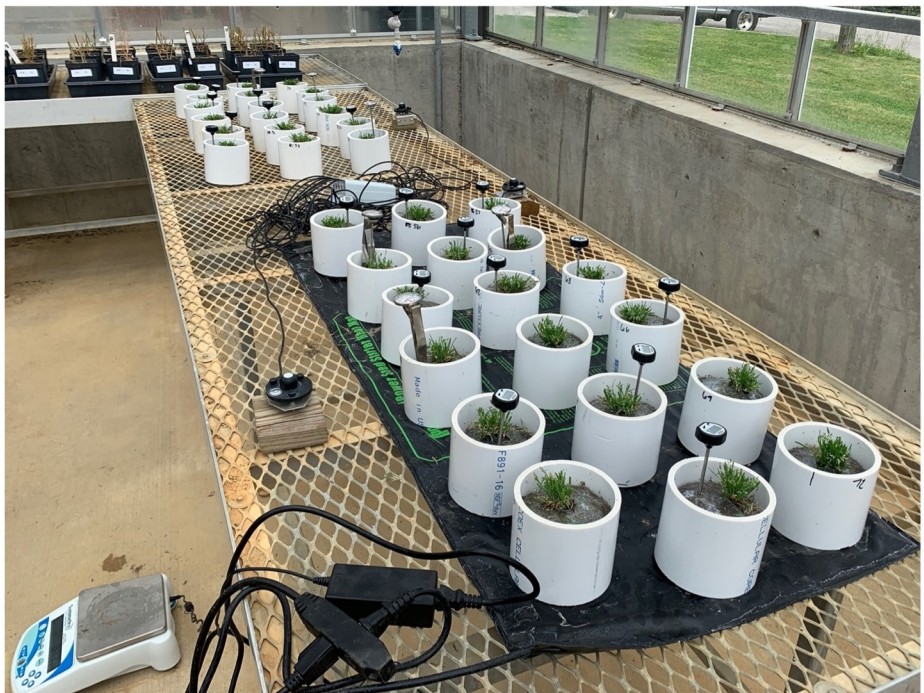

**Fig 3. *Poa annua* plants harvested on 21 April 2021 in Knoxville, TN and wrapped in 0.15 mm polyethylene plastic to prevent soil water evaporation.** Plants were subjected to either ambient or elevated soil temperature via a heating mat. Within each soil temperature treatment, water was either completely withheld or added to meet transpirational loss.

minutes. Volumetric water content (VWC) of soil was recorded at approximately a 5.0 cm depth on the day of mortality, or for those that survived, at the conclusion of the study by inserting a soil moisture sensor (ML3 ThetaProbe; Delta -T Devices, Cambridge, UK) into the center of each core and recording a single reading.

Visually rated mortality data were subjected to life cycle analysis in Prism (Prism 9 for Mac, Graph-Pad software, La Jolla, CA) using the Kaplan-Meier estimate of survival probabilities log rank test with $P < 0.0001$ determining differences in mortality.

## Disease susceptibility evaluation

A field experiment was conducted at the East Tennessee AgResearch and Education Center (Knoxville, TN) from 27 May to 28 October 2021 to assess the impacts of fungicide application on *P. annua* mortality. The experimental site was a mixed stand of *P. annua* and zoysiagrass (*Zoysia matrella* Merr. cv. Trinity) atop a Sequatchie silt loam soil with a pH of 6.2 and 2.9% organic matter. The area was mown twice per week at a 1.3 cm height of cut and watered four times per week via automatic irrigation. Slow-release fertilizer (Gal-Xe$^{ONE®}$; Simplot, Boise, ID) was applied to the entire area on 22 April 2021 to deliver 195 kg N ha$^{-1}$ yr$^{-1}$. On 25 May 2021, the experimental area was treated with quinclorac (Drive XLR8; BASF Corporation, Research Triangle Park, NC) at 0.84 kg ha$^{-1}$ + methylated seed oil (0.5% v/v) to control summer annual weeds.

The experiment was arranged as a randomized complete block design with 10 replications of two treatments applied to 1 m$^2$ plots. Treatments included either no fungicide application or fungicide mixtures including active ingredients from Fungicide Resistance Action Committee groups #2, 3, 5, 7, 11, and 21 intended to manage outbreaks of dollar spot, anthracnose,

*Pythium* blight, and brown patch (S2 Table). Treatments were applied every 14 days from 26 May to 13 October 2021. Every two weeks, experimental units were visually assessed for *P. annua* mortality. An F-score analysis of climatic data associated with mortality was conducted using the same methods and packages as those used in the life cycle monitoring experiment.

## Results & discussion

### Life cycle observation

In four of the 10 micro-environments (#2, #6, #7, and #9) all ten monitored *P. annua* plants survived the summer season, whereas in the other six micro-environments, all ten monitored plants senesced during the observational period. The ability of *P. annua* to survive year-round in certain environments in this study aligns with reports of perennial ecotypes [6–30].

Surviving plants exhibited a range of morphological characteristics. Plants from locations two and six were fine-textured, displayed lateral growth, and produced inflorescences periodically throughout summer (S1 Fig). Plants in locations seven and nine were coarse-textured with an upright growth habit and lacked inflorescences. Micro-environments conducive to *P. annua* survival were those where conditions were likely such that water was either not a limiting factor or where fungicides were applied. Analysis of environmental factors at the time of observed plant mortality corroborated these observations and indicated that sum precipitation, maximum soil temperature, and likelihood of pathogenic infection were significant factors related to plant death (Fig 4).

### Soil temperature evaluation

Analysis of variance revealed only a significant soil temperature treatment effect in photochemical efficiency data. Therefore, data were pooled across experimental runs and ecotypes. Considering that ecotypes in this experiment were harvested from distinct microenvironments where we observed mortality or continued survival in our observational study, the

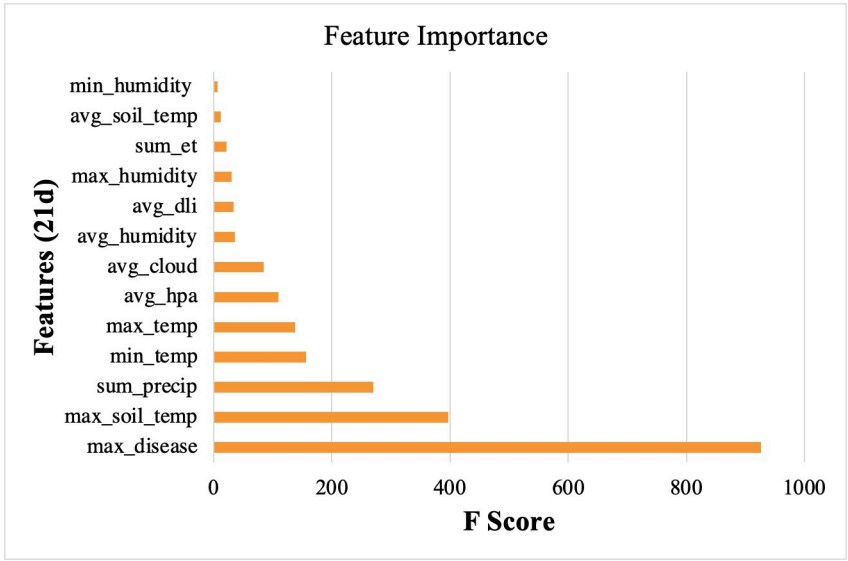

**Fig 4. F-score analysis conducted in Python of climatic data related to death events of *Poa annua* monitored for length of life from May to October 2020 in ten micro-environments in Knoxville, TN.**

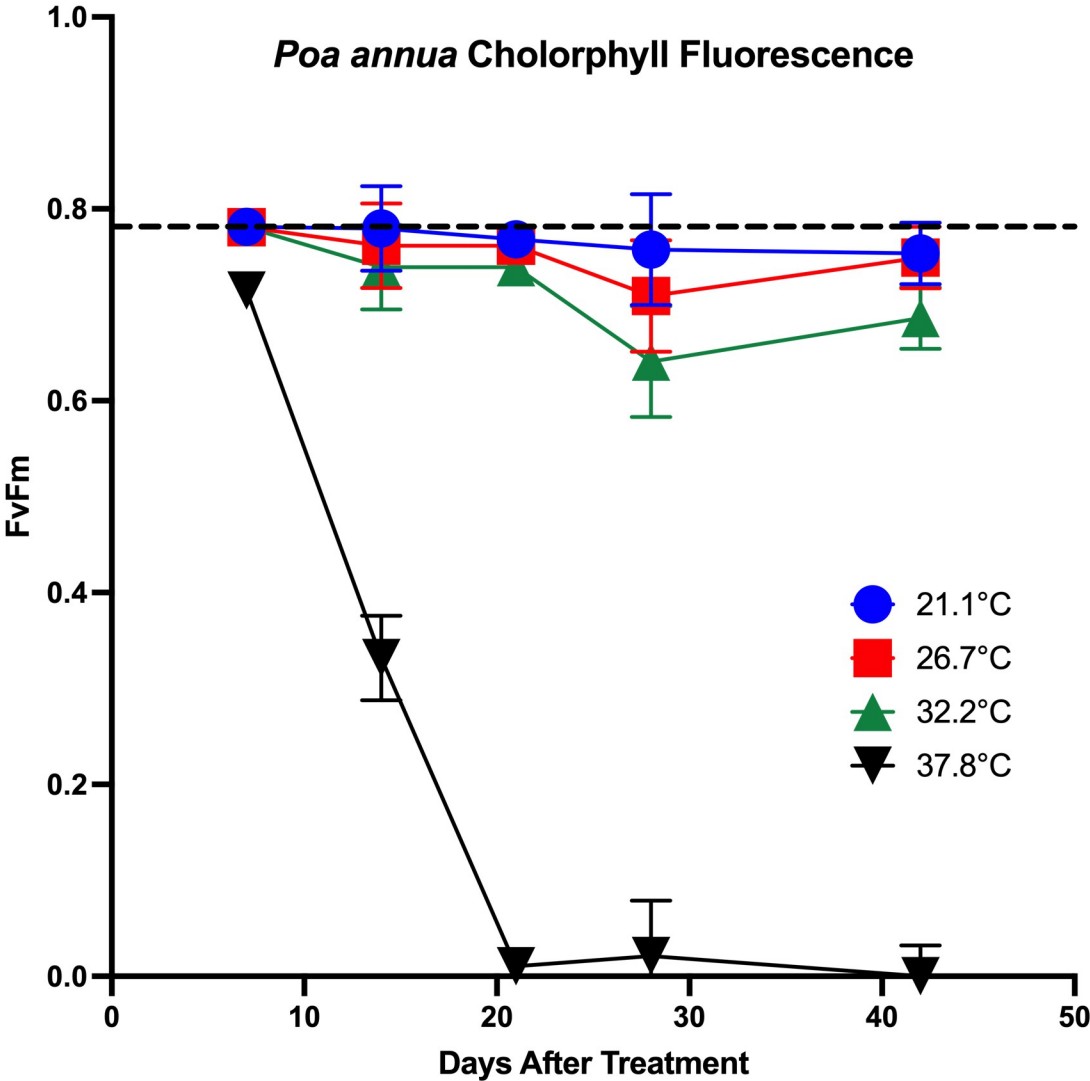

**Fig 5. Photochemical efficiency was evaluated by measuring chlorophyll fluorescence (Fv/Fm) twice per *Poa annua* plant using a fluorometer.** Measurements were taken once per week from plants subjected to soil temperatures of 21.1, 26.7, 32.2, or 37.8°C imposed via water baths. The dotted line indicates optimal chlorophyll fluorescence of 0.79.

lack of an ecotype effect in photochemical efficiency provides evidence that *P. annua* mortality is dictated by environmental conditions rather than programmed senescence common of annual plants.

*Poa annua* plants survived 28 days of exposure to soil temperatures of 21.1, 26.7, or 32.2°C and senesced when subjected to 37.8°C. Plants subjected to the 21.1°C soil temperature remained healthy for the duration of the experiment as photochemical efficiency was near optimum (Fv/Fm of 0.79), ranging from 0.75 to 0.78 (Fig 5). Conversely, plants subjected to the 37.8°C soil temperature exhibited drastically reduced photochemical efficiency from 7 DAT until the end of the study. *Poa annua* plants exposed to the 26.7 or 32.2°C soil temperature treatments did not senesce but were negatively affected by prolonged exposure to elevated soil temperatures; both treatments numerically reduced photochemical efficiency from 21 to 42 days after treatment (DAT). For plants subjected to the 37.8°C soil temperature,

photochemical efficiency was drastically reduced from 0.71 at 7 DAT to 0.33 by 14 DAT, remaining ≤ 0.02 thereafter, indicating death.

Only the 37.8˚C soil temperature treatment was lethal to *P. annua* plants with mortality first noted 12 DAT. By 17 DAT, 35 of 36 *P. annua* plants exposed to the 37.8˚C soil temperature had died. Reductions in *P. annua* plant health without death, except in extreme cases, when soil temperature is elevated supports other published reports [32, 51]. For example, exposure of 115 *P. annua* selections to conditions of 47˚C at 100% relative humidity in a growth chamber for six hours did not cause mortality, however 77% of plants were negatively affected (< 50% recovery) [32]. Similarly, *P. annua* subjected to day/night air temperatures of 20/15, 30/25, or 40/35˚C for eight days in growth chambers resulted in physiological damage only from the supraoptimal temperature of 40/35˚C, although no plants senesced [51].

Survival of *P. annua* in these glasshouse experiments agreed with observations made in our life cycle monitoring experiment in the field. For example, peak soil temperature across all ten micro-environments was 30.8˚C (The Earthstream Platform; mesur.io, Yanceyville, NC) in 2020 and average daily air temperature did not exceed 28.7˚C. Given that some *P. annua* plants in the observational experiment senesced without soil temperatures exceeding 32.2˚C, we concluded that elevated soil temperature alone does not result in *P. annua* death.

## Soil water evaluation

A log-rank test detected significant (*P* <0.0001) differences in survival probability among treatments. Survival probability for plants subjected to the well-watered + ambient soil temperature treatment was 100% throughout the experiment (Fig 6). No treatment resulted in a survival probability of 0% during the tested time frame. However, across all treatments, 27 of 72 tested *P. annua* plants did senesce. The lowest *P. annua* survival probability was observed for the full drought + elevated soil temperature treatment; Probability of survival was not reduced from 100% until 19 DAT and remained above 50% through 40 DAT. Comparatively, survival probability for plants exposed to the well-watered + elevated soil temperature and full drought + ambient temperature treatments was 100% through 34 DAT and remained above 70% through 42 DAT.

The ability of *P. annua* to survive an extended period with low water availability is documented [50]. These researchers did not observe *P. annua* senescence when plants were maintained in a sand-based medium with VWC from 4 to 12% for 95 days. While VWC treatments tested are similar to those achieved via drought stress treatment (6.6 to 8.4%) in this study, air and soil temperature were not published in their report [50]. Although a combination of full drought and elevated soil temperature was lethal to some *P. annua* plants in our experiment, it is telling that the probability of *P. annua* survival was still 25% after 42 days of continuous exposure to conditions that rarely occur in the field. Moreover, no other treatment combination in this experiment reduced the probability of *P. annua* survival to < 70% during the study. These responses indicate that interactions of soil moisture and temperature are not lethal to *P. annua* except in extreme situations atypical of managed urban grasslands.

## Disease susceptibility evaluation

*Poa annua* plants in plots not treated with fungicide did not survive the summer period. High individual plant mortality was observed in non-treated areas on 24 June 2021 with some plots completely devoid of *P. annua* beginning on 8 July 2021 (Fig 7). By 4 August 2021, no *P. annua* plants were alive in any non-treated plots. Conversely, in plots treated with fungicide, some *P. annua* plants survived to the termination of the experiment.

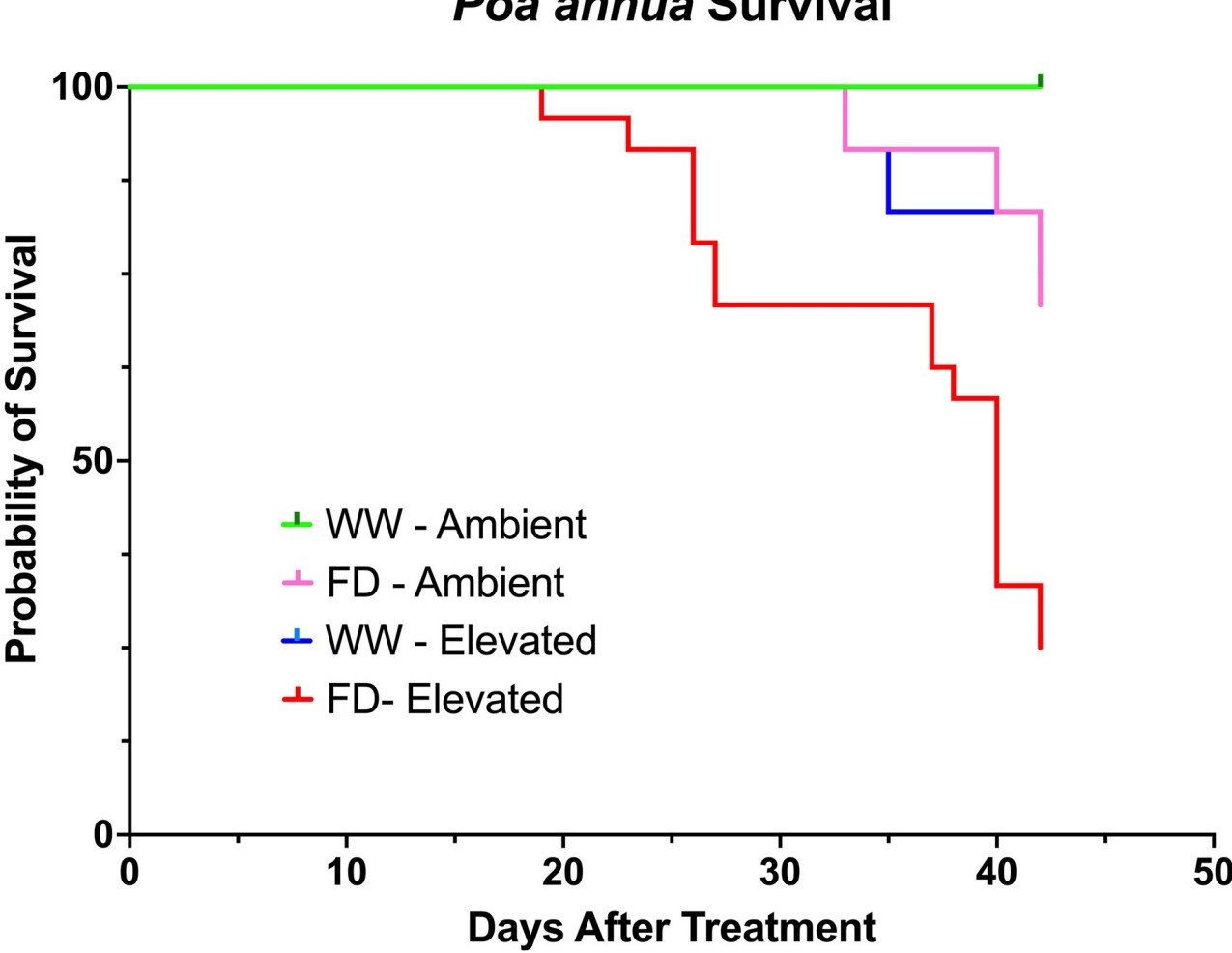

**Fig 6. Plot of survival probabilities for *Poa annua* plants maintained at ambient or elevated soil temperatures and subjected to either well-watered or full drought conditions for 42 days.** Probabilities were generated using the Kaplan-Meier estimate of survival function in Prism.

The range of the anthracnose caused by *C. cereale* severity (Fig 7) in this experiment suggests that increases in zoysiagrass cover throughout the monitoring period (and a concomitant decline in *P. annua* plant size) may have affected our ability to document *P. annua* during summer. For example, zoysiagrass cover across the experimental area at study initiation was only 63%, whereas by 18 August 2021, zoysiagrass cover reached 88%. Zoysiagrass is a C₄ species adapted to the summer season in Knoxville, TN and *P. annua* is a C₃ species that is at a disadvantage during summer. Therefore, we surmised that zoysiagrass outcompeted *P. annua* for resources and caused a natural population dynamic shift. Inter-species competition with common bermudagrass (*Cynodon dactylon* Pers.) has limited *P. annua* persistence in summer months previously [8]. As zoysiagrass grew denser, *P. annua* became more difficult to assess via visual ratings. Without the presence of an inflorescence to clearly identify *P. annua* plants, it is possible plants were not identified, but persisted throughout the season (Fig 8). For example, fungicide treated plots scored as not containing *P. annua* on 4 August contained multi-tillered plants in autumn when temperatures cooled and suggest that the roots and underground shoots of the plants had survived the summer and grew rapidly when favorable environmental

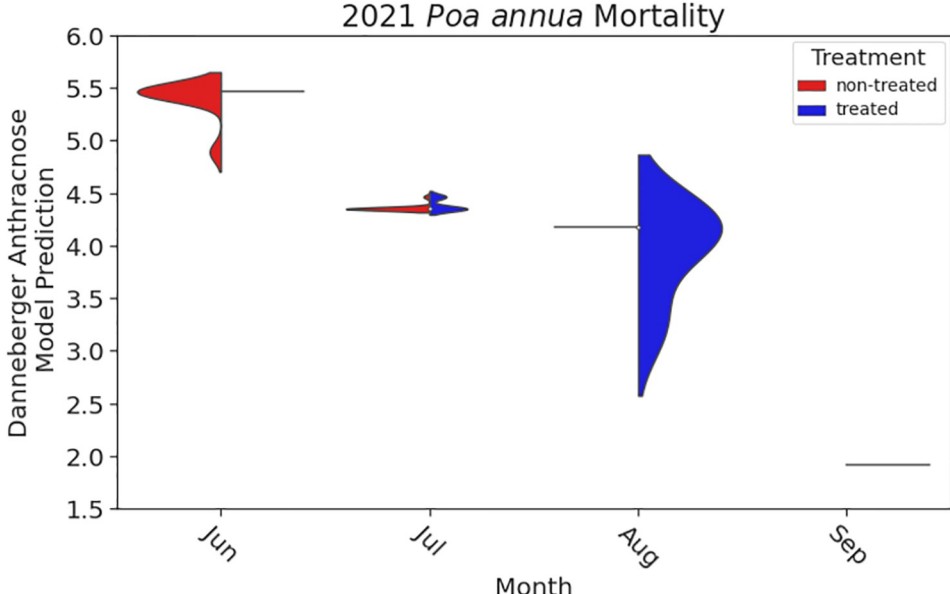

**Fig 7. *Poa annua* mortality following either no fungicide treatment or fungicide treatment every 14 days from 27 May to 28 October 2021.** The y-axis represents anthracnose (*Colletotrichum cereale*) severity index calculated using the Danneberger model [53], where an anthracnose severity index of 2 is the minimum conditions for infection. The x-axis represents time, indicated by month of year. Within each month, *Poa annua* mortality is split by fungicide treatment with vertical space equating to the range of anthracnose severity across the month and horizontal space equating to the smoothed number of senescence observations corresponding to each treatment. The figure was created in Python using 'matplotlib' (v. 3.5.0; https://matplotlib.org/) and seaborn (v. 0.11.2).

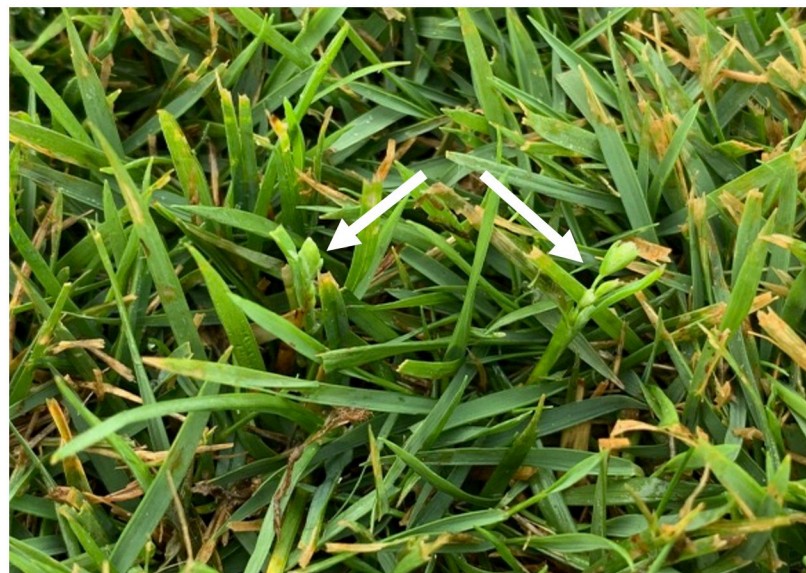

**Fig 8. *Poa annua* plants with inflorescences on 4 August 2021 following biweekly fungicide applications in a zoysiagrass (*Zoysia matrella* Merr. cv. 'Trinity') urban grassland located in Knoxville, TN.** Without the presence of inflorescences, plants would likely be difficult to identify.

conditions for growth were restored. This observation points to potential selection for ecotypes with an increased root to shoot ratio [67], which could support year-long survival via stress reduction or capitalization of resource acquisition compared with newly germinated seedlings.

Interestingly, several *P. annua* plants maintained vigorous growth throughout the experiment, producing inflorescences as early as 4 August 2021 (Fig 8). *Poa annua* emergence models developed in Knoxville, TN indicate peak emergence typically occurs between the 40th and 43rd week of the calendar year when mean seven-day soil temperature lowers to $\leq 18.9$ºC and seven-day rainfall accumulation is $\geq 12.7$ mm [68]. Observation of *P. annua* inflorescence production in only the 32nd week of the year when soil temperatures were $> 24$ ºC and average weekly rainfall was $< 4.75$ mm (The Earthstream Platform; mesur.io, Yanceyville, NC) supports our assertion that observed plants were not newly germinated but had survived the summer season. Comparatively early inflorescence production aligns with findings of differing vernalization requirements and time to reproductive maturity among *P. annua* ecotypes [21, 22, 67].

Results of this experiment indicate that fungal pathogens are the primary cause of *P. annua* mortality in maintained urban grasslands. Although not all plants in fungicide treated plots survived or were observable, *P. annua* plants treated with fungicide lived markedly longer than those not treated with fungicide (Fig 7). F-score analysis of climatic data at the time of plant mortality revealed risk of anthracnose and brown patch infection as significant factors associated with mortality (Fig 9). On the first date mortality was documented in non-treated plots (24 June 2020), anthracnose acervuli were present on necrotic foliage.

Summer decline of *P. annua* via fungal infection is thoroughly discussed in pathology literature [52, 58, 69]. Fungicide use on *P. annua* managed as a desirable species is a readily accepted management practice for summer survival [70, 71]. Some reports indicate *P. annua* cannot be commercially released until disease resistance is incorporated into the species because plants

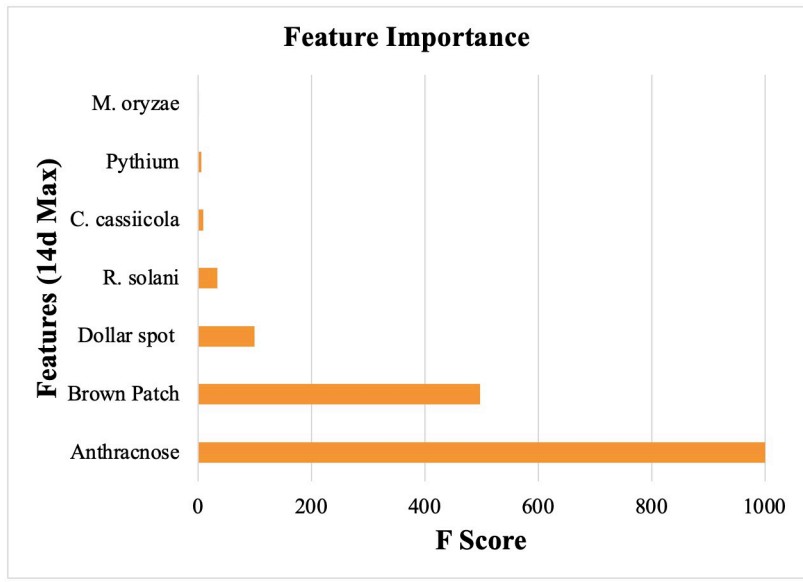

**Fig 9. F-score analysis conducted in Python of climatic data related to death events of *Poa annua* plants monitored for length of life from May to October 2021 in a zoysiagrass (*Zoysia matrella* Merr., cv. Trinity) fairway.** Plants were either left without treatment or treated with fungicide mixtures intended to control dollar spot (*Clarireedia* spp.), anthracnose (*Colletotrichum cereale*), Pythium blight (*Pythium* spp.), and brown patch (*Rhizoctonia solani*) every 14 days.

are so highly susceptible to pathogen attack [8]. Therefore, we recommend that susceptibility to fungal pathogens be considered when assessing the life cycle of *P. annua*.

## Conclusions

Results of these experiments suggest *P. annua* perishes from fungal infection, which may be exacerbated by environmental and anthropogenic stresses during summer months [56, 57, 59]. *Poa annua* seems to persist unless environmental conditions are unfavorable, often presenting as a polycarpic plant rather than succumbing to programmed senescence. This environmentally driven response is similar to that of other perennial $C_3$ grass species such as Kentucky bluegrass (*Poa pratensis* L.) or tall fescue [*Schedonorus arundinaceus* (Schreb.) Dumort]. On the contrary, annual species are monocarpic plants senescing after a single reproductive cycle regardless of environment [3]. Our results show that *P. annua* does not meet the definition of an annual species. Observations made in this study and via an exhaustive review of peer-reviewed literature [60] present little evidence supporting an inherently annual life cycle in *P. annua*. We contend the epithet "annua" is a misnomer according to its current interpretation. Although "annual" can be associated with life cycle, "occurring once every year" is also a definition of the word [72]. Thus, the species epithet "annua" may have been awarded to *P. annua* to mark a yearly observation such as growth at the same location or yearly inflorescence production rather than of an annual life cycle.

Early taxonomic descriptions of *P. annua* provide no evidence of life cycle study at the time of naming, indicating the species name has been misconstrued in modern times [60]. The shift in present understanding of *P. annua* as a perennial rather than the long purported annual species has major implications for plant management. Therefore, a more appropriate and descriptive name may be *P. typica* or *P. vulgari*, meaning 'typical' and 'common' in Latin, respectively. These epithets preclude misinformation surrounding life cycle and indicate the species prevalence given that *P. annua* is found on all seven continents in a myriad of climates.

## Supporting information

**S1 Fig. *Poa annua* L. plants of wide morphological variation in texture and growth habit observed to survive from 11 May to 27 October 2020 in Knoxville, TN.** Images taken on 4 August 2020 where A = a fine-texture, laterally growing plant with inflorescence in micro-environment two; B = a fine-texture, laterally growing plant in micro-environment six; C = an upright growing, coarse textured plant in micro-environment seven; and D = an upright growing, coarse textured plant in micro-environment nine.
(TIF)

**S1 Table. *Poa annua* life cycle experiment workflow.**
(DOCX)

**S2 Table. Fungicide applications made between May and October 2021 to control diseases common to *Poa annua*.**
(DOCX)

## Acknowledgments

Authors thank Drs. Kellie Walters and John Zobel for their assistance in developing methods. Authors would also like to recognize the contributions made by Javier Vargas, Gregory Breeden, and Benjamin Pritchard in experiment set up and management.

## Author Contributions

**Conceptualization:** Devon E. Carroll, Brandon J. Horvath, Robert N. Trigiano, Avat Shekoofa, Thomas C. Mueller, James T. Brosnan.

**Data curation:** Devon E. Carroll, Michael Prorock.

**Formal analysis:** Devon E. Carroll, Michael Prorock, James T. Brosnan.

**Investigation:** Devon E. Carroll, Avat Shekoofa.

**Methodology:** Devon E. Carroll, Brandon J. Horvath, Michael Prorock, Robert N. Trigiano, Avat Shekoofa, Thomas C. Mueller, James T. Brosnan.

**Project administration:** Devon E. Carroll, James T. Brosnan.

**Resources:** James T. Brosnan.

**Software:** Michael Prorock.

**Supervision:** Brandon J. Horvath, Avat Shekoofa, James T. Brosnan.

**Visualization:** Michael Prorock.

**Writing – original draft:** Devon E. Carroll.

**Writing – review & editing:** Devon E. Carroll, Brandon J. Horvath, Michael Prorock, Robert N. Trigiano, Avat Shekoofa, Thomas C. Mueller, James T. Brosnan.

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
