## [Decision Letter · Decision Letter 0]

19 Aug 2022

PONE-D-22-14930Poa annua: An annual species?PLOS ONE

Dear Dr. Brosnan

Thank you for submitting your manuscript to PLOS ONE. After careful consideration, we feel that it has merit but does not fully meet PLOS ONE’s publication criteria as it currently stands. Therefore, we invite you to submit a revised version of the manuscript that addresses the points raised during the review process. Please submit your revised manuscript by Oct 03 2022 11:59PM. If you will need more time than this to complete your revisions, please reply to this message or contact the journal office at plosone@plos.org. Please include the following items when submitting your revised manuscript:A rebuttal letter that responds to each point raised by the academic editor and reviewer(s). You should upload this letter as a separate file labeled 'Response to Reviewers'.A marked-up copy of your manuscript that highlights changes made to the original version. You should upload this as a separate file labeled 'Revised Manuscript with Track Changes'.An unmarked version of your revised paper without tracked changes. You should upload this as a separate file labeled 'Manuscript'.If applicable, we recommend that you deposit your laboratory protocols in protocols.io to enhance the reproducibility of your results. Protocols.io assigns your protocol its own identifier (DOI) so that it can be cited independently in the future. For instructions see: https://journals.plos.org/plosone/s/submission-guidelines#loc-laboratory-protocols. Additionally, PLOS ONE offers an option for publishing peer-reviewed Lab Protocol articles, which describe protocols hosted on protocols.io. Read more information on sharing protocols at https://plos.org/protocols?utm_medium=editorial-email&utm_source=authorletters&utm_campaign=protocols.

We look forward to receiving your revised manuscript.

Kind regards,

Mehmet Cengiz Baloglu

Academic Editor

PLOS ONE

Journal Requirements:

Additional Editor Comments:

Dear Dr. Brosnan,

Thank you for submitting your manuscript for review to the Plos One. After careful consideration, we feel that your manuscript will likely be suitable for publication if it is revised to address the points below. Therefore, my decision is "Minor Revision."

We invite you to submit a revised version of the manuscript that addresses the following points and reviewers’ and my critics:

Our reviewers reported that there are some typos and minor issues. Our referees also made some corrections and suggestions. They agreed that the paper is well-written and experiments were well constructed. However our reviewers have concerns about requirements of identification of random and fixed variables. In addition, the workflow, dates, treatments and statistical analysis of the study should be explained more clearly. I also agree with all these opinions. Corrections and advice should be performed in the revised version of the manuscript. Please include a rebuttal letter that responds to each point brought up by the academic editor and all reviewers.

There are several issues, which need the attention of the authors to make it a sound study;

I strongly believe that the extensive criticism from reviewers will help the authors to make it suitable for the publication of the manuscript in Plos One.

Yours sincerely,

Reviewers' comments:

Reviewer's Responses to Questions

**Comments to the Author**

1. Is the manuscript technically sound, and do the data support the conclusions?

Reviewer #1: Yes

Reviewer #2: Yes

2. Has the statistical analysis been performed appropriately and rigorously? 

Reviewer #1: I Don't Know

Reviewer #2: Yes

3. Have the authors made all data underlying the findings in their manuscript fully available?

Reviewer #1: Yes

Reviewer #2: Yes

4. Is the manuscript presented in an intelligible fashion and written in standard English?

Reviewer #1: Yes

Reviewer #2: Yes

5. Review Comments to the Author

Reviewer #1: The paper is well-written and experiments were well constructed.

With respect to the analysis of the results, identification of random and fixed variables needs to be indicated.

Some literature expresses a difference in vernalization (late versus early flowering) may be an indicator for the annual versus perennial biotypes (continuous versus restricted flowering times) and that this can impact within plant dynamics (e.g. root:shoot ratio) and potentially survival, due to reduction in stress? The experiments found in this paper make similar conclusions to previous work, that growth environment will not necessarily impact persistence, and that there is a greater range of ecotypes within this species and that end-use management will likely select for the best biotype for the given use.

Reviewer #2: The manuscript by Devon E. Carroll et al. searches for an answer to whether Poa annua is an annual or perennial plant. In order to determine the environmental factors lethal to Poa annua, a series of glasshouse or field experiments were conducted. However, some issues need to be addressed to improve the quality of the manuscript.

1. The workflow, dates and treatments of the study were given in a complex fiction in the text. The workflow and its details should be explained more clearly in the text. A figure with workflow, dates, and practices can be added to make it easier for the reader to catch up.

2. In the soil temperature evaluation study, plants from micro-environment #1 are included to eliminate the effect of the ecotype (lines 319-325). Why were plants from micro-environment #1 not included in the study in which soil water was evaluated, as in the soil temperature study?

3. On line 310, “Micro-environments conducive to P. annua survival were those where water was either not a limiting factor or where fungicides were applied.” sentence is passed, but such an application is not mentioned in the methods of the Life cycle study.

4. As a result of the study, it was stated that the diseases caused by microorganisms are the primary reason Poa annum senescence, but it seems problematic that the result, which is said to be the strongest, is in the experimental set with the most problematic observations. There is a need for improvement in this issue. Additional discussion can help.

5. It is not mentioned in the text on what basis statistical calculations of microorganisms are made and how data is obtained. It seems that it would be better if the researcher wrote this subject more descriptively.

6. The resolution and quality of Figure-5 and Figure-6 need to be increased

6. PLOS authors have the option to publish the peer review history of their article (what does this mean?). If published, this will include your full peer review and any attached files.

Reviewer #1: No

Reviewer #2: No

---

## [Editor Report · Decision Letter 1]

28 Aug 2022

Poa annua: An annual species?

PONE-D-22-14930R1

Dear Dr. Brosnan,

We’re pleased to inform you that your manuscript has been judged scientifically suitable for publication and will be formally accepted for publication once it meets all outstanding technical requirements.

Kind regards,

Mehmet Cengiz Baloglu

Academic Editor

PLOS ONE

Additional Editor Comments (optional):

Dear Dr. Brosnan,

Thank you for re-submitting your revised manuscript for review to the Plos One. After careful consideration, the revised manuscript has been corrected based on our reviewer’s comments. So, the presented results are sufficient for a scientific paper and manuscript has been also met with the journal criteria.

Point by point, I reviewed all the corrections. Each comment was addressed carefully by the authors.

I am pleased to inform you that your manuscript has been deemed suitable for publication in Plos One.

Yours sincerely,

Prof. Dr. Mehmet Cengiz BALOGLU, Ph.D.

Academic Editor

Plos One.
---

## [Editor Report · Acceptance letter]

31 Aug 2022

PONE-D-22-14930R1 

*Poa annua*: An annual species? 

Dear Dr. Brosnan:

I'm pleased to inform you that your manuscript has been deemed suitable for publication in PLOS ONE. Congratulations! Your manuscript is now with our production department. 

Kind regards, 

on behalf of

Prof. Dr. Mehmet Cengiz Baloglu 

Academic Editor

PLOS ONE